# Reverse mutational scanning of SARS-CoV-2 spike BA.2.86 identifies epitopes contributing to immune escape from polyclonal sera

Najat Bdeir[1], Tatjana Lüddecke[1], Henrike Maaß [1], Stefan Schmelz[2], Ulfert Rand [3], Henning Jacobsen[1], Kristin Metzdorf[1], Upasana Kulkarni[1], Anne Cossmann[4], Metodi V. Stankov [4], Markus Hoffmann [5,6], Stefan Pöhlmann [5,6], Wulf Blankenfeldt [2,7], Alexandra Dopfer-Jablonka [4,8], Georg M. N. Behrens [4,8,9,10] & Luka Čičin-Šain [1,8,9,10] ✉

The recently detected Omicron BA.2.86 lineage contains more than 30 amino acid mutations relative to BA.2. BA.2.86 and its JN.1 derivative evade neutralization by serum antibodies of fully vaccinated individuals. In this study, we elucidate epitopes driving the immune escape of BA.2.86 and JN.1 via pseudovirus neutralization. Here we generate 33 BA.2.86 mutants, each reverting a single mutation back to BA.2. We use this library in an approach that we call reverse mutational scanning to define distinct neutralization titers against each epitope. Mutations within the receptor binding domain at K356T, V483Δ, and to a lesser extent N460K, A484K, and F486P enhance immune escape. Interestingly, 16insMPLF within the spike N-terminal domain and P621S within S1/S2 also significantly contribute to antibody escape of BA.2.86. Upon XBB.1.5 booster vaccination, neutralization titers against JN.1 and BA.2.86 improve considerably, and residual immune escape is driven by 16insMPLF, N460K, E554K, and to a lesser extent P621S, and A484K.

The emergence of new SARS-CoV-2 virus lineages continues to be a critical aspect of the ongoing epidemic viral spread. Among these lineages, BA.2.86, also known as Pirola, has garnered recent attention owing to its significant antigenic shift away from the prevailing XBB sub-lineage[1,2]. The earliest detection of BA.2.86 was in late July 2023 in Denmark[3–5]. By mid-august, it had been detected within several countries and WHO had classified it as a variant of interest[4–6]. An outbreak of BA.2.86 recorded in the United Kingdom with a high attack rate (86.6%) within an elderly care home demonstrated the transmissibility of this lineage[7]. At present, the extent of disease severity exerted by BA.2.86 is unclear, but its derivative sub-variant JN.1 is on track to become the globally dominant SARS-CoV-2 lineage.

[1]Department of Viral Immunology, Helmholtz Centre for Infection Research, Braunschweig, Germany. [2]Department Structure and Function of Proteins, Helmholtz Centre for Infection Research Braunschweig, Braunschweig, Germany. [3]DSMZ- German Collection of Microorganisms and Cell Cultures, Braunschweig, Germany. [4]Department of Rheumatology and Immunology, Hannover Medical School, Hannover, Germany. [5]Infection Biology Unit, German Primate Center – Leibniz Institute for Primate Research, Göttingen, Germany. [6]Faculty of Biology and Psychology, University Göttingen, Göttingen, Germany. [7]Institute for Biochemistry, Biotechnology and Bioinformatics, Technische Universität Braunschweig, Braunschweig, Germany. [8]German Center for Infection Research, partner site Hannover-Braunschweig, Hannover, Germany. [9]Centre for Individualized Infection Medicine, a joint venture of HZI and Hannover Medical School, Hannover, Germany. [10]These authors contributed equally: Georg M. N. Behrens, Luka Čičin-Šain.
✉e-mail: Luka.Cicin-Sain@helmholtz-hzi.de

The viral spike (S) protein mediates SARS-CoV-2 host cell entry through a multistep process. The initial step involves binding of the S protein to angiotensin converting enzyme-2 receptors (ACE2). This engagement is followed by S protein cleavage by host cell proteases, enabling the S protein to drive fusion of the viral envelope with cellular membranes[8]. The S1 domain of the S protein entails an N-terminal domain (NTD) with somewhat unclear functions, and the receptor-binding domain (RBD), which directly binds to ACE2 and is the major target for neutralizing antibodies[8–10]. The transmembrane S2 domain drives viral fusion with the host cell membrane, which facilitates the release of viral genetic material into the cytoplasm, and therefore plays an important role in infection. BA.2.86 harbors more than 30 mutations relative to BA.2, encompassing 13 mutations in NTD, 14 in the RBD, and 7 within the pre S1/S2 and S2 domain[11]. Furthermore, several BA.2.86 descendants have been identified, including BA.2.86.1 (defining mutation ORF1a:K1973R), JN.1 (L455S), JN.2 (ORF1a:Y621C), JN.3 (ORF1a:T2087I), and BA.2.86.2 (ORF7a:E22D)[2].

The alarming number of BA.2.86 spike mutations has prompted several efforts to characterize the antibody immune escape potential of this lineage. Recent studies demonstrate reduced pseudo-virus neutralization of BA.2.86 and JN.1 in comparison to BA.2 and B.1 strains and that vaccination with the monovalent BNT162b2 XBB.1.5 adapted vaccine significantly enhances neutralization of BA.2.86 pseudo virus by serum antibodies[12–14]. However, the contribution of single mutations to the immune escape of BA.2.86 remains unclear. Mutational scanning approaches, where libraries of viruses with single amino acid mutations in the spike protein are compared to the wild-type virus are powerful tools for the identification of epitopes recognized by monoclonal antibodies[15–17], but polyclonal serum antibodies recognize numerous epitopes simultaneously and redundantly. Therefore, mutating one out of 33 epitopes on an ancestral background may only marginally decrease the serum neutralizing activity if some among the remaining 32 epitopes are recognized by independent antibody clones. To overcome this limitation, we cloned a library of 33 reversion mutants on the BA.2.86 background, each harboring a single mutation reverting the position back to the amino acid in BA.2. This approach allowed us to observe a robust increase in neutralizing activity whenever an immunologically relevant epitope was reintroduced in the spike. We tested this library of BA.2.86 mutants against serum samples collected from a cohort of 30 healthcare workers, before and after vaccination with the BNT162b2 XBB.1.5 adapted vaccine. Our data showed that mutations ins16MPLF, K356T, N460K, V483Δ, A484K, F486P and S621P distributed across NTD, RBD, and S1/S2 domains, contribute to the immune escape of BA.2.86. Additionally, we show that vaccination with the BNT162b2 XBB.1.5 adapted vaccine increases substantially the neutralization titers against both BA.2.86 and the more recent BA.2.86.1.1 (JN.1) descendant, and that the immune escape of JN.1 is more pronounced than that of BA.2.86 before and after XBB.1.5 booster vaccination. Moreover, we demonstrate that the deletion of the MPLF insertion at position 16, reversions N460K and K554E, and to lesser extent the reinsertion of the tyrosine residue at position 144, as well as S621P and K484A improve neutralization of BA.2.86 upon the XBB.1.5 booster shot.

## Results

### BA.2.86 spike protein harbors a substantial amount of mutations within all domains

The analysis of the BA.2.86 spike sequence (specifically: hCoV-19/Denmark/DCGC-647694/2023, EPI_ISL_18114953) revealed 33 mutations relative to BA.2 spike (Fig. 1). These include 13 mutations within the NTD, 14 mutations in the RBD, and 6 mutations within the S2 and pre S1/S2 domain. Of these mutations, there are five deletions (H69Δ, V70Δ, Y144Δ, N211Δ, and V483Δ) and one insertion after V16 (V16insMPLF). Mutations that have been previously identified in other variants of interest are R21T (B.1.617), H69Δ/V70Δ (B.1.1.7/Alpha),

Y144Δ (XBB.1.5; EG.5.1; BA.1), R158G (B.1.617.2/Delta), E484K (B.1.351/Beta; P.1/Gamma) and P681R (B.1.617.2/Delta). Additionally, BA.2.86 harbors several mutations, which were rarely reported (V445H, N450D, N481K, V483Δ; and E554K)[18–20]. Among these mutations ins16MPLF, ΔY144, F157S, R158G, H245N, A264D are located within the NTD antigenic supersite and may contribute to immune escape[21]. Additionally, several mutations within the RBD of BA.2.86 have been associated with antibody resistance including K356T, A484K, and N450D[9,22,23], while several other mutations R493Q, F486P, N460K, and V483Δ may alter ACE2 interactions[24,25]. Hence, BA.2.86 contains a plethora of mutations within the spike protein, which may alter key properties of this virus in receptor binding and neutralizing antibody escape. To visualize the position of mutations in the spike protein of BA.2.86 with respect to BA.2, we used AlphaFold2/AlphaFold-Multimer[26,27] to construct a structural model of the spike trimer of BA.2.86. The model, which was obtained in a closed state with respect to the conformation of the RBD, shows that mutations with respect to BA.2 are spread over the distal part of the protein but otherwise do not cluster at specific positions (Fig. 1B, C).

### BA.2.86 and JN.1 efficiently escape antibody neutralization

To assess the immune escape of the BA.2.86 and BA.2.86.1.1 (JN.1) lineages, we employed pseudo-virus particles (pp) in neutralization assays. For comparison, we also included particles harboring the spike protein of XBB.1.5 (XBB.1.5pp), Wuhan-Hu-01 (WTpp), BA.1 (BA.1pp), and BA.2 (BA.2pp) (Fig. 2A, B). We found that plasma obtained from a cohort of at least double boostered individuals neutralized BA.1pp and BA.2pp with a 3-fold reduced efficiency as compared to the index WTpp. However, the inhibition of BA.2.86pp and XBB.1.5pp was 57- and 48-fold reduced, respectively (Fig. 2A). Antibody escape of JN.1pp was even more pronounced, with a ~141-fold reduction relative to WTpp (Fig. 2A). Plasma acquired post vaccination with the XBB.1.5-adapted mRNA vaccine neutralized XBB.1.5pp and BA.2.86pp, with almost comparable efficiency, whereby the mean neutralization titer was 9-fold and 11-fold lower than WTpp, respectively, while JN.1pp neutralization was 27 fold lower (Fig. 2B). Collectively, BA.2.86pp and JN.1pp escaped neutralization by antibodies induced upon primary vaccination series and boosters with immunogens predating XBB lineages, whereby this escape was more pronounced in JN.1pp. However, vaccination with the XBB.1.5 adapted vaccine boosted the neutralizing titers against both variants and reduced the gap in neutralization efficiency between them and Omicron BA.2. Additionally, the neutralization efficiency of JN.1pp was ~2.4 fold reduced relative to BA.2.86pp, before and after XBB.1.5 booster vaccination.

### Mutations ins16MPLF, K356T, N460K, V483Δ, A484K, F486P, and P621S contribute to BA.2.86 neutralizing antibody escape

To investigate the effect of individual mutations within BA.2.86 on post-vaccination neutralizing antibody escape, we cloned a comprehensive library of 33 individual BA.2.86 mutants. Each of them contains a single reversion relative to the amino acid sequence of BA.2, while retaining the rest of the sequence as in BA.2.86. Geometric means of pseudo-virus neutralization titers (PVNT50) against BA.2.86pp were ~18-fold lower than against BA.2pp prior to vaccination with the XBB.1.5 vaccine (Fig. 3A). Hence, we tested which mutations decreased the gap between neutralization of BA.2pp and BA.2.86pp. Our data showed that among the BA.2.86pp mutants with single mutations within the NTD, only the insMPLF16Δpp reduced the gap to BA.2pp to a mere 4-fold reduction (Fig. 3A). The remaining NTDpp single mutants did not significantly contribute to neutralizing antibody escape, because their neutralization titers were comparable to that of BA.2.86pp (Fig. 3B). Moreover, we have shown in Zhang et al. that insMPLF16Δ leads to a reduction of infectivity in VeroE6 cells[28]. Hence, this unique NTD mutation is involved in both immune evasion from sera antibodies and infectivity. While the N-terminus of BA.2.86 spike protein is modeled

with lesser confidence than the core of the structure (Supplementary Fig. S1A–C), it is interesting that the N-terminal 16MPLF insertion is predicted to interact with a crevice in the N-terminal domain (NTD; Supplementary Fig. S1B, C). This is reminiscent of SARS-CoV spike protein[29], albeit here, the N-terminus is yet more extended and anchored via a disulfide bridge to the core of the NTD (Supplementary

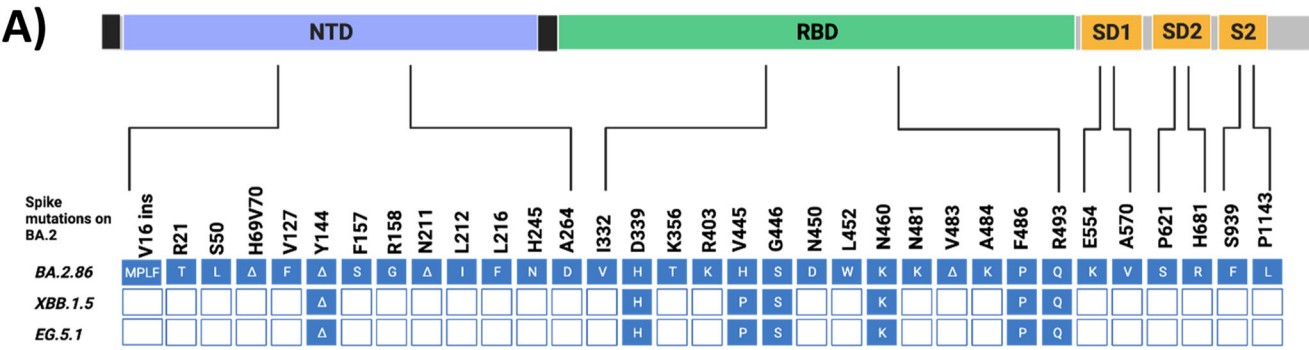

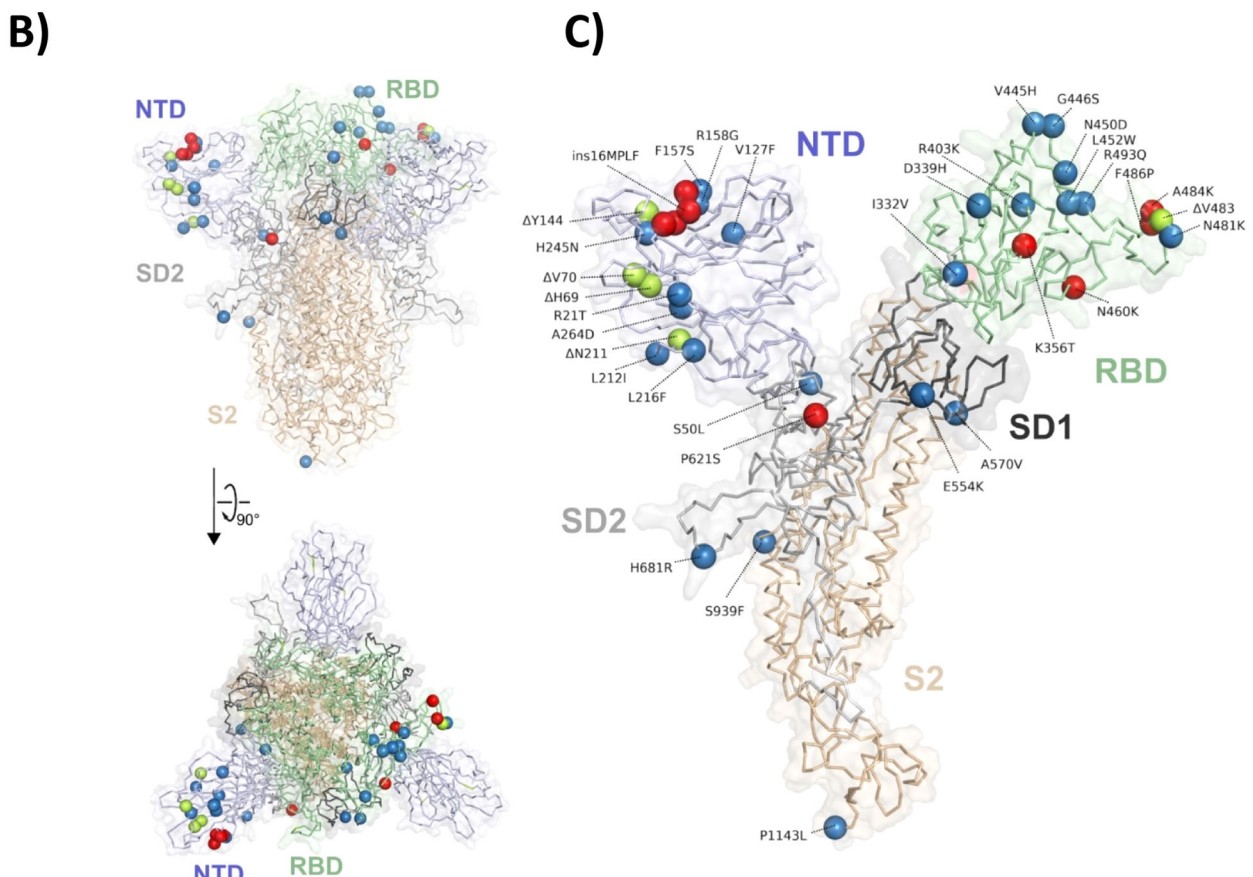

**Fig. 1 | Overview of BA.2.86 lineage specific spike protein mutations relative to BA.2. A** Schematic representation of the SARS-CoV-2 spike domains and amino acid changes indicated for BA.2.86, and shared by XBB.1.5, and EG.5.1 compared to the spike of BA.2. Further mutations within XBB.1.5 and EG.5.1 relative to BA.2 are not shown, as only sites mutated within BA.2.86 are represented. N-terminal domain (NTD, blue), receptor binding domain (RBD, green), Subdomains 1 and 2 (SD1 and SD2, orange), S2 subunit (orange). Created in BioRender. Bdeir, N. (2024) BioRender.com/m05k539 and in the references add: Bdeir, N. (2024). Figure A. Created in BioRender. BioRender.com/m05k539. **B** Model of the trimeric spike protein of BA.2.86, calculated with AlphaFold2/AlphaFold-Multimer[27,45]. The N-terminal secretion signal (15 residues) and the C-terminal membrane-anchoring sequence (112 residues) were omitted from calculations, leading to 3372 residues in the final model. Domains have been colored according to panel **A**, and one of five independently calculated models is shown. Spheres represent the location of mutations with respect to the spike protein of BA.2. The positions of deletions are colored in green, red spheres indicate mutations that lead to enhanced immune escape of BA.2.86, other mutations are shown in blue. For clarity, mutations are only shown in one chain of the spike trimer. **C** Magnified view of one trimer extracted from the model shown in **B** and shown in the same orientation.

## A)   Before XBB.1.5 booster vaccination

| Response rate: | 100% | 100% | 100% | 60% | 63% | 37% |
| --- | --- | --- | --- | --- | --- | --- |
| Geometric mean titer: | 5375 | 1847 | 1646 | 113 | 94 | 38 |
| Fold reduction (vs.WT): | — | 3 | 3 | 48 | 57 | 141 |

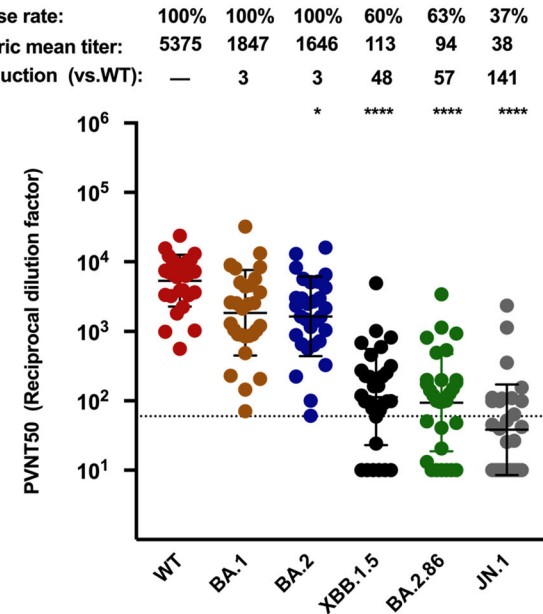

## B)   After XBB.1.5 booster vaccination

| Response rate: | 100% | 100% | 100% | 96.6% | 96.6% | 76.6% |
| --- | --- | --- | --- | --- | --- | --- |
| Geometric mean titer: | 13517 | 8569 | 7969 | 1467 | 1280 | 493 |
| Fold change (vs.WT): | — | 2 | 2 | 9 | 11 | 27 |

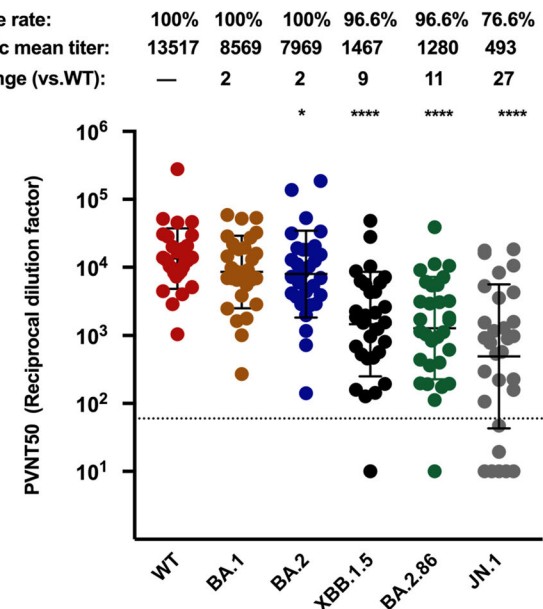

**Fig. 2 | BA.2.86 and JN.1 efficiently evade neutralization in double boostered individuals, but the adapted XBB.1.5 vaccine booster enhances protection against both. A** Particles pseudo-typed with the indicated S proteins were pre-incubated for one hour at 37 °C with plasma dilutions from double boostered health care workers (biological replicates n = 30) (2A) or with plasma dilutions following vaccination with an adapted XBB.1.5 booster (biological replicates *n* = 30) (2B). Pseudo-virus neutralization titer 50 (PVNT50) was calculated using the least squares fit using a variable slope, using a four-parameter nonlinear regression model and values were plotted as geometric mean. Geometric mean standard deviation bars are shown in black. The lower limit of confidence (LLOC) was set at a

PVNT50 of 50. Non responders are defined as individuals below 60 (dashed line). All PVNT50 below 10 are set at 10 for visualization purposes. The assay was performed in technical duplicates and with negative controls to assess the virus input of each used pseudo-virus in the absence of plasma antibodies. Percentage of positive responders, geometric means, and fold change neutralization over WT-Wuhan$_{pp}$ are shown on top. Friedman two sided non parametric paired test (ns, $p > 0.05$; *$p \le 0.05$; **$p \le 0.01$; ***$p \le 0.001$). Percentage of positive responders, geometric means, and fold change neutralization over WT-Wuhan$_{pp}$ are shown on top. Source data are provided as a Source Data file.

Fig. S1D). Structural modeling from Colson et al.[30] also suggest that insMPLF16 masks a V-shaped electronegative zone within the NTD which might allow an advantage to stabilize the virus on target cells. It may also induce long range conformational rearrangements affecting the RBD and potentially RBD-ACE2 interactions.

The neutralization capacity of serum samples against epitopes in the RBD of BA.2.86$_{pp}$ was significantly affected prior to the XBB.1.5 booster by the mutations K356T and ΔV483, which had a 6 or 7 fold reduction in neutralization titer relative to BA.2, respectively (Fig. 4A, B). Additionally, our results showed that K460N$_{pp}$, K484A$_{pp}$, and P486F$_{pp}$ had a 6-, 7-, and 8-fold reduced neutralization efficiency, respectively, relative to BA.2$_{pp}$, and thus much less than the 18-fold reduction observed in BA.2.86$_{pp}$. While the latter results did not raise to statistical significance over BA.2.86$_{pp}$ (Fig. 4B), they indicated an improved neutralization in the presence of these parental epitopes. In contrast, the remaining mutations within the RBD of BA.2.86 did not enhance serum neutralization capacity.

In addition to the aforementioned RBD and NTD BA.2.86 mutants, we explored the contribution of mutations within the S1/S2 and S2 regions to antibody evasion. We report a significant increase in neutralization efficiency for BA.2.86 S621P$_{pp}$, whereby neutralization efficiency of BA.2.86 S621P$_{pp}$ was 4-fold lower than that of BA.2$_{pp}$ and hence, neutralization efficiency is significantly improved in comparison to that of BA.2.86$_{pp}$ (Fig. 5A, B) In contrast, BA.2.86 K554E$_{pp}$, V570A$_{pp}$, R681H$_{pp}$, F939S$_{pp}$, L1143P$_{pp}$ all showed comparable neutralization sensitivity as compared to BA.2.86$_{pp}$ (Fig. 5A, B).

Interestingly, the impact of BA.2.86 mutations on neutralization escape following vaccination with the adapted XBB.1.5 immunogen

revealed that the mutants insMPLF16Δ$_{pp}$, K554E$_{pp}$, and K460N$_{pp}$ significantly recovered the neutralization efficiency of plasma samples above BA.2.86$_{pp}$ and to levels comparable to that of BA.2 neutralization (Fig. 6). An increase in neutralization efficiency after XBB.1.5 booster vaccination was observed for BA.2.86 mutants S621P$_{pp}$, ins144Y$_{pp}$ and K484A$_{pp}$, but it did not raise to statistical significance, (Fig. 6). Further information on the post vaccination neutralization titers for each mutant pseudovirus is provided in supplementary data (Supplementary Fig. S2 and Supplementary Data 2). Since neutralization efficiency of JN.1$_{pp}$ was merely 2.4 fold reduced relative to BA.2.86$_{pp}$ after the XBB.1.5 booster vaccination, our data may argue that these positions may be relevant for the residual immune escape of JN.1 upon the XBB.1.5 booster as well.

We previously reported that pseudovirus infectivity of Vero cells was significantly perturbed in Δins16MPLF, K403R, D450N, Q493R, and S621P[28]. Since a reduction in infectivity may reflect an increase of dysfunctional virions in a stock, which would act as sponges for neutralizing antibodies and compete with infectious particles[31], we performed a digital droplet PCR (ddPCR) of VSV-N stocks, thus defining the genome copy number relative to infectious virus titers. For all tested mutants, these numbers were clustering between those of BA.2 and BA2.86 (Supplementary Fig. S3). Therefore, our results all but exclude the possibility that the neutralization titers were affected by the infectivity of the individual epitope mutants.

## Discussion

This work provides to our knowledge the first example of a reverse mutational scanning strategy for the identification of epitopes that

## Neutralisation titers for NTD specific BA.2.86pp mutants pre XBB.1.5 adapted vaccine

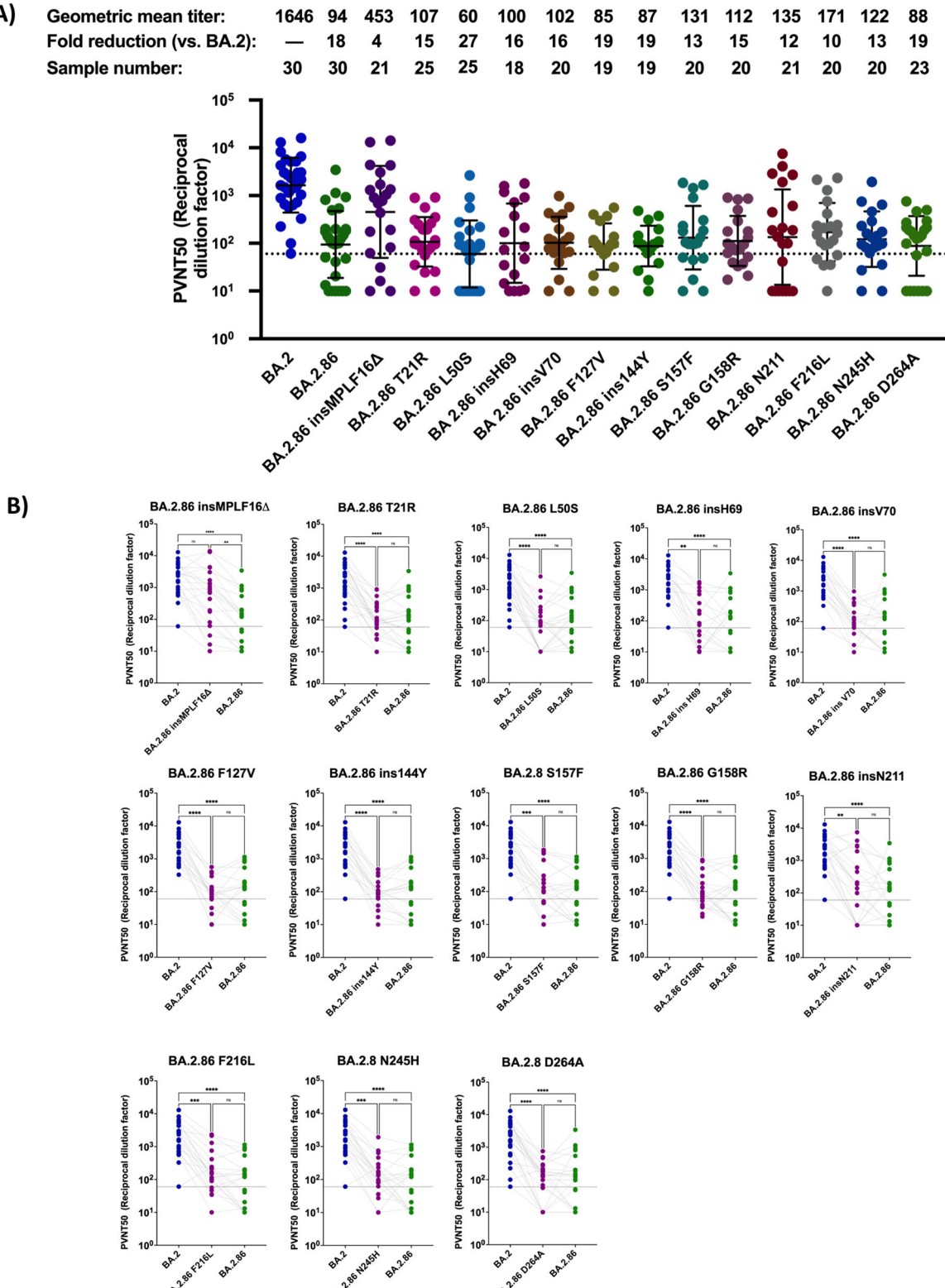

**A)**

| | BA.2 | BA.2.86 | BA.2.86 insMPLF16Δ | BA.2.86 T21R | BA.2.86 L50S | BA.2.86 insH69 | BA.2.86 insV70 | BA.2.86 F127V | BA.2.86 ins144Y | BA.2.86 S157F | BA.2.86 G158R | BA.2.86 N211 | BA.2.86 F216L | BA.2.86 N245H | BA.2.86 D264A |
|---|---|---|---|---|---|---|---|---|---|---|---|---|---|---|---|
| Geometric mean titer: | 1646 | 94 | 453 | 107 | 60 | 100 | 102 | 85 | 87 | 131 | 112 | 135 | 171 | 122 | 88 |
| Fold reduction (vs. BA.2): | — | 18 | 4 | 15 | 27 | 16 | 16 | 19 | 19 | 13 | 15 | 12 | 10 | 13 | 19 |
| Sample number: | 30 | 30 | 21 | 25 | 25 | 18 | 20 | 19 | 19 | 20 | 20 | 21 | 20 | 20 | 23 |

contribute to immune escape from vaccine-induced immunity. Strategies based on mutational scanning of the SARS-CoV-2 spike are not new. We and others have shown that such approaches can be used to generate libraries containing individual mutations present in the Omicron, but not in the ancestral variants, and thus identify epitopes that escape recognition by monoclonal antibodies[16,17] and polyclonal sera[15,16,31]. A very comprehensive mutational scanning based on the XBB.1.5 spike has been recently performed by the Bloom lab to introduce 9000 theoretical mutations on the XBB.1.5 background and thus predict future potential escape mutations[15]. However, forward mutational scanning cannot predict the emergence of lineages with big evolutionary jumps, such as the Omicron BA.1 and BA.2.86 variants,

**Fig. 3 | Mapping mutations in the NTD for effects of BA.2.86 neutralization efficiency in double boostered individuals. A** Neutralization assessment for particles pseudo-typed with mutations within the NTD of BA.2.86. Each mutant shown contains a single mutation reverting the amino acid mutation in BA.2.86 to the corresponding amino acid within BA.2. Particles pseudo-typed with the indicated S proteins were preincubated for one hour at 37 °C with plasma dilutions from double boostered health care workers with non-adapted immunogens. The numbers of biological replicates corresponding to individual plasma samples are shown on top. Pseudo-virus neutralization titer 50 (PVNT50) was calculated using the least squares fit using a variable slope, using a four-parameter nonlinear regression model. The lower limit of confidence (LLOC) was set at a PVNT50 of 50.

Non responders are defined as individuals below 60 (dashed line) All PVNT50 below 10 are set at 10 for visualization purposes. The assay was performed in technical duplicates and with negative controls to assess the virus input of each used pseudovirus in the absence of plasma antibodies. Percentage of positive responders, geometric means, and fold change neutralization over WT-Wuhan$_{pp}$ are shown on top. Geometric mean standard deviation bars are shown in black. **B** Individual neutralization data for particles pseudo-typed with mutations within the NTD of BA.2.86. Statistical significance was assessed by Friedman two sided non parametric paired test (ns, $p > 0.05$; *$p \leq 0.05$; **$p \leq 0.01$; ***$p \leq 0.001$). Source data are provided as a Source Data file.

where more than 30 mutations were observed simultaneously, with no intermittent stages that are known. Moreover, such approaches are not ideal for the analysis of polyclonal sera neutralizing potential, where redundant epitope recognition may result in virus neutralization even if immunologically relevant epitopes are mutated. Indeed, a forward mutational scanning approach that we have employed in a previous study to assess Omicron epitope mutations on a Wuhan background showed only modest (<two-fold) reductions in neutralization titers[31]. By reverse mutational scanning, we provide here a loss-of-function genetic approach, allowing the identification of up to fourfold differences in titers. Thus, we were able to identify escape epitopes in polyclonal responses to antigens with many simultaneous mutations, defining the collection of epitopes that contribute to immune escape from vaccine-induced immunity by BA.2.86 and its derivative JN.1 lineage. This approach may also be rapidly deployed for subsequent lineages with big evolutionary jumps that may emerge in the future.

The emergence of BA.2.86 harboring more than 30 mutations relative to BA.2 was reminiscent of the Omicron appearance in 2021 and raised concerns regarding its antibody escape potential[12,13,32]. A number of these mutations (K356, V445, G446, N450, L452, and P621) were also observed in omicron sub-lineages within immunocompromised patients. This may indicate that reduced immune functions within some individuals may be a source of highly mutated SARS-CoV2 lineages[33,34]. BA.2.86 has also evolved several descendants including JN.1 which harbors three mutations in non S-proteins and a hallmark S455L mutation in the spike protein[2]. In plasma from a cohort that was at least double boostered, we observed that the S455L mutation reduces the neutralization efficiency relative to BA.2.86 by a factor of ~2.5, but that the XBB.1.5 adapted vaccine booster increases neutralization efficiency against both lineages roughly ~12 to 13 fold, in line with data reported in Stankov et al.[35] and Wang et al.[13]. Moreover, neutralization of BA.2.86$_{pp}$ and JN.1$_{pp}$ were merely 1.2-fold and 3 fold lower than the neutralization of Omicron XBB.1.5 respectively. Nevertheless, their neutralization titers remain 6- and 16 fold lower than that of BA.2 following vaccination respectively.

Several mutations within BA.2.86 significantly increased the neutralizing antibody escape prior to vaccination with the adapted XBB.1.5 vaccine. We found that the NTD mutation ins16MPLF significantly affected neutralization sensitivity, and its reversion resulted in a 4-fold increase in neutralization titers relative to BA.2.86. While this region of the NTD is disordered in published structures of the SARS-CoV-2 spike protein, indicating high intrinsic flexibility, the MPLF insertion is somewhat reminiscent of the SARS-CoV spike protein, where the N-terminus is yet more extended and anchored via a disulfide bridge to the core of the NTD (Supplementary Fig. S1D). It should be noted that several NTD-binding neutralizing antibodies have been identified in the past[36,37], indicating that mutations in this domain may indeed interfere with the immune system's capacity to recognize the virus. Even more interestingly, we show that the deletion of the MPLF insertion at position 16 significantly increases neutralization efficiency after the adapted XBB.1.5 vaccination to BA.2 comparable levels. This mutation is located within the NTD antigenic supersite, which is a key

target for NTD specific neutralizing antibodies[21]. Moreover, in silico structural modeling of BA.2.86 performed by Colson et al. indicates that the MPLF insertion may mask a V-shaped electronegative zone within the NTD, which is an unprecedented phenotype in SARS-CoV-2. This zone may stabilize the virus onto target cells and may induce some long-range conformational changes which affect the RBD with potential consequences on RBD-ACE2 interactions[30]. However, our independent in silico analysis of this region of the NTD structure argues that the changes at the N-terminal tip cannot be predicted with a high degree of confidence. Therefore, the actual effects of this mutation on the NTD structure may only be confirmed by empirical analysis in cryoelectron microscopy or similar approaches.

We also found that the mutation K356T within the RBD plays a contributing role to BA.2.86 escape of neutralization by polyclonal sera, whereby it lead to a six-fold reduction in neutralization efficiency relative to BA.2$_{pp}$, in comparison to an 18 fold reduction in the case of BA.2.86$_{pp}$. This reduction in neutralization efficiency might be attributed to the steric hindrance caused by the introduction of an additional glycosylation site[38]. Similarly, we show that mutations N460K, V483Δ, A484K, and F486P within the RBD may enhance neutralizing antibody escape. This is in line with reports from Wang et al. which show that mutation N460K and F486P shared in XBB.1.5 and EG.5.1 cause resistance to class 1 and 2 monoclonal antibodies (mAb)[39]. Structural modeling has shown that the mutation N460K, which was first identified in BA.2.75, disrupts a hydrogen bond formed between the RBD and a class 1 mAb (VH3-53)[40] and a study by Wang et al. demonstrated that the mutation A484K within BA.2.86 reduced the neutralizing activity of a subset of class 3 mAbs[13]. The mutation V483Δ has seldom been reported in circulating strains. Full spike mutational scanning of BA.2.86 postulated that V483Δ may contribute to antibody escape but experimental evidence for this has been lacking[15]. We also show that the mutation P621S in the S1/S2 domain of BA.2.86 contributed to significant neutralization escape and this phenotype, to our knowledge has not been demonstrated previously. In sum, we have identified several mutations that have significantly contributed to immune escape in our cohort. However, we cannot exclude that additional mutations may result in immune escape in individuals whose repertoire differs from our cohort of double boostered individuals. This requires additional studies in cohorts of elderly people or those with immune deficiencies. In parallel to our present study, we have conducted a more thorough investigation to explore the effects of BA.2.86 specific mutations on infectivity and cell entry, including ACE-2 binding and protease usage[28]. We observed that lung cell entry of BA.2.86 is ACE2 dependent and is markedly more efficient than that driven by the S proteins of other omicron sublineages. To obtain insights into which amino acids are critical to this phenotype, we explored the infectivity of every pseudoviral mutant on Vero and Calu-3 lung cells. This revealed that BA.2.86 specific mutations at NTD residue L50 and RBD residue T356 specifically promote lung cell entry and bear no relevance for Vero cells' entry. Several mutations affected cell entry into Vero cells, up to a sevenfold reduction in infectivity.

We have shown previously that mutant viruses with a hundred fold impaired infectiveness display a quantifiable reduction in

**Neutralisation titers for RBD specific BA.2.86pp mutants pre XBB.1.5 adapted vaccine**

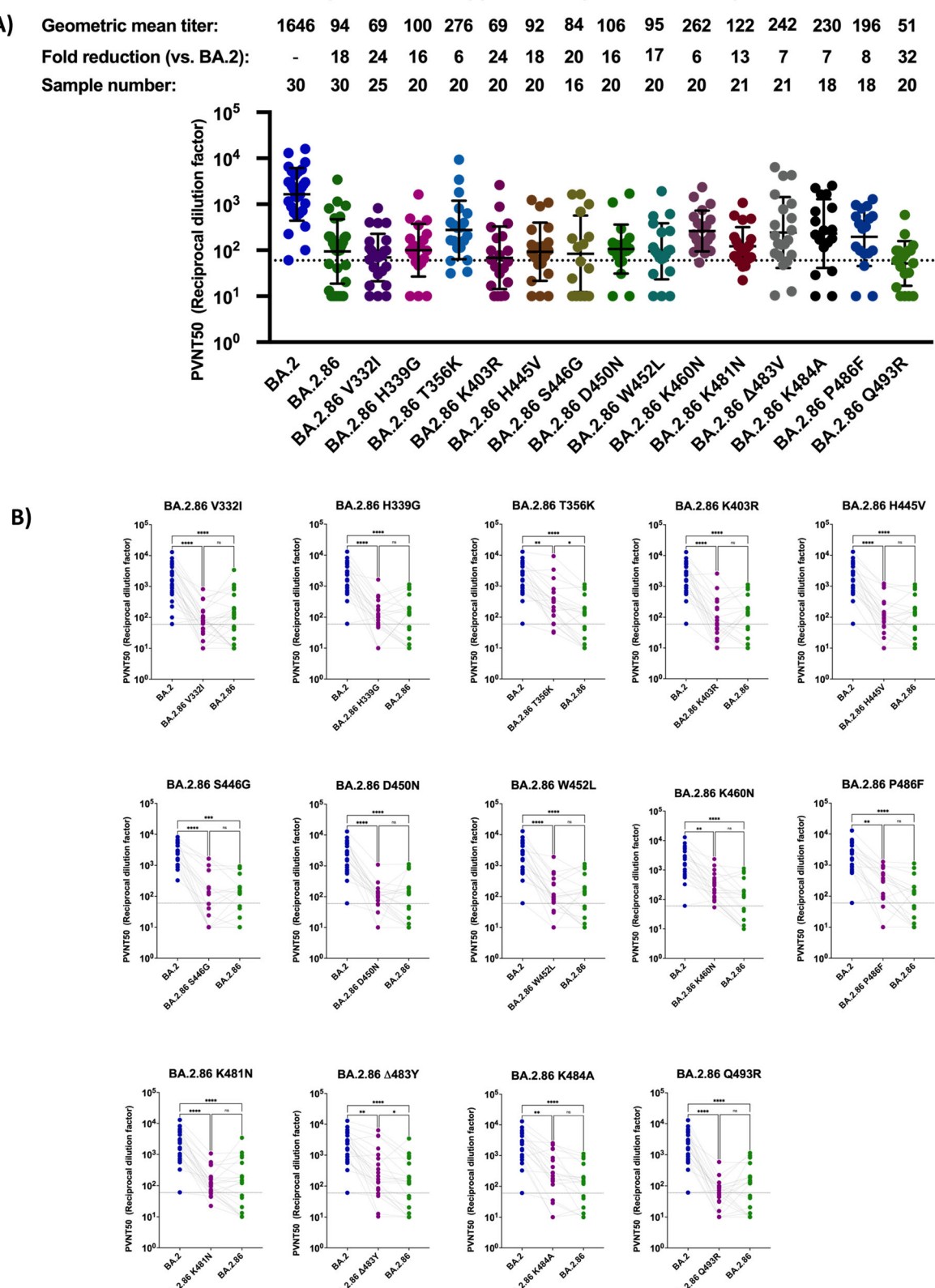

neutralization serum titers due to the preponderance of non-infectious virus particles in the virus stock that act as antibody sponges in the neutralization assay[31]. Thus, we checked if the pseudovirus stocks showing significant differences in the cell entry assay into Vero cells[28] or in the neutralization assay had more non-infectious particles (Supplementary Fig. S3), but we observed no substantial differences

relative to the parental BA.2 or the index BA.2.86 strain. Hence, effects of individual mutations on Vero cell infectivity was unlikely to have affected our neutralization results.

It is also important to acknowledge some pertinent limitations relating to our study. We have utilized the well-established pseudovirus system to assess the contribution of single mutations to the

**Fig. 4 | Mapping mutations in the RBD for effects of BA.2.86 neutralization efficiency in double boostered individuals. A** Neutralization assessment for particles pseudo-typed with mutations within the RBD of BA.2.86. Each mutant shown contains a single mutation reverting the amino acid mutation in BA.2.86 to the corresponding amino acid within BA.2. Particles pseudo-typed with the indicated S proteins were pre-incubated with serum dilutions from immunized health care workers for one hour at 37 °C. The numbers of biological replicates corresponding to individual plasma samples are shown on top. The lower limit of confidence (LLOC) was set at a PVNT50 of 50. Non responders are defined as individuals below 60 (dashed line). Non responders are defined as individuals

below this threshold. All PVNT50 below 10 are set at 10 for visualization purposes. The assay was with negative controls to assess the virus input of each used pseudo-virus in the absence of plasma antibodies. Percentage of positive responders, geometric means, and fold change neutralization over BA.2$_{pp}$ are shown on top. Geometric mean standard deviation bars are shown in black. **B** Individual neutralization data for particles pseudo-typed with mutations within the RBD of BA.2.86. Statistical significance was assessed by Friedman two sided non parametric paired test (ns, $p > 0.05$; *$p ≤ 0.05$; **$p ≤ 0.01$; ***$p ≤ 0.001$). Source data are provided as a Source Data file.

antibody escape potential of BA.2.86. While formal verification would require the use of authentic SARS-CoV-2 with spike mutations introduced by reverse genetics, neutralization titers in pseudo-viral and authentic virus setups have been shown to be comparable due to the immunodominance of spike over other structural elements[41–43]. An additional limitation in our study is the lack of information regarding hybrid immunity within our cohort, whereby some participants may have experienced a prior unrecorded infection with XBB sublineages, which may have elicited a humoral immune response similar to vaccination. However, only 4 individuals have a recorded infection in 2023, when the XBB.1.5 lineage was present at high levels.

In sum, BA.2.86 and its JN.1 descendent efficiently escape neutralization by polyclonal serum antibodies of at least double boostered individuals, and our data argue that this is due to mutations at positions N460K, V483Δ, A484K, F486P, K356T, P621S, and ins16MPLF. We also observed that the S455L mutation provides a 2-fold decrease in neutralization titers over BA.2.86. However, neutralization titers of JN.1 and BA.2.86 were appreciably improved by the XBB.1.5 vaccine booster, whereby JN.1 has a mere 2.5 reduction relative to BA.2.86. This may argue that residual immune escape of both lineages may rely on the shared epitopes at positions ins16MPLF, E554K, N460K, and to lesser extent A484K and Y144Δ.

## Methods
### Cell lines
All cell lines were maintained at 37 °C and 5% $CO_2$ in a humified environment. 293T (Human, kidney) and VeroE6 (African green monkey, kidney) cells were cultured in Dulbecco's Modified Eagle Medium (DMEM, ThermoFisher Scientific) supplemented with 5% fetal bovine serum (FBS, ThermoFisher Scientific) and 100 U/ml penicillin and 0.1 mg/ml Streptomycin (PAN-Biotec). Both cell lines were used to a maximum passage of 30. For seeding and sub-cultivation, cells were washed with phosphate buffered saline (PBS, PAN-Biotec) and then incubated with trypsin/EDTA (PAN-Biotec) until cell detachment. Cell lines were routinely tested for mycoplasma. The origin and catalog numbers of cells are shown in table 1 in Supplementary Data 3. Transfection of 293T cells for the production of pseudoviruses was carried out by calcium phosphate transfection.

### Plasmids
The plasmid pCG1_SARS-2-Sdel18 (Codon-optimized) encoding the spike protein of the Wuhan-Hu-1 SARS-CoV-2 has been previously reported[8]. The pCG1_SARS-2-Sdel18 BA.1 and BA.2 expression plasmids are previously reported[31] and based on isolate hCoV-19/Botswana/R40B58_BHP_3321001245/2021 (GISAID Accession ID: EPI_ISL_6640919) and isolate hCoV-19/England/PHEC-4G0AFZF7/2021 (GISAID Accession ID: EPI_ISL_8738174) respectively. The pCG1_SARS-2-Sdel18 XBB expression plasmid was generated by Gibson assembly based on the expression vector for the spike of Omicron BA.2 and site directed mutagenesis was done to generate XBB.1.5. Expression plasmids pCAGGS-DsRed and pCG1-SARS-2-SDel18 BA.2.86 (based on the isolate hCoV-19/Denmark/DCGC-647694/2023, EPI_ISL_18114953) were kindly provided by the Laboratory of Stefan Pöhlmann (Supplementary Table 3 in Supplementary Data 3). Site directed mutagenesis (Q5®

High-Fidelity 2X Master Mix, New England BioLabs) was utilized for the generation of the SARS-CoV-2 spike BA.2.86 expression plasmid library containing single point mutations back to BA.2 spike. Transformation was carried out using NEB® 10-beta Competent E. coli from New England BioLabs (catalog: C3019H). Primers are listed in Table 2 in Supplementary Data 3.

### Production and titration of pseudo-viruses
Production of pseudo-viruses was performed according to published protocol[44]. In brief, 293T cells were seeded in 6 well plates at a confluency of 70%. The next day, cells were transfected with expression plasmids for pCG1-SARS-2-SΔ18 WT, pCG1-SARS-2-SΔ18 BA.2.86, pCG1-SARS-2-SΔ18 BA2, pCG1-SARS-2-SΔ18 BA.2.86 XBB or pCG1-SARS-2-SΔ18 BA.2.86 single point mutants. At 24 h post transfection, cells were incubated for 1 h with a replication deficient VSV (VSV*ΔG) expressing enhanced green fluorescent protein (eGFP) and firefly luciferase at an MOI of 3. VSV*ΔG was kindly provided by the lab of Gert Zimmer. Subsequently, cells were washed with phosphate buffered saline (PBS) and incubated with anti-VSV-G antibody (mouse hybridoma supernatant from CRL-2700; ATCC) in order to neutralize residual input virus. At 12 h post infection, supernatants were harvested and cleared from cell debris by centrifugation and stored at −80 °C for later use. For titration, aliquots of the pseudo-virus were serially diluted threefold in duplicates, starting with a 1:10 dilution in a 96-well plate, totaling eleven dilutions. The last column served as a control with cells but no pseudo-virus. Confluent VeroE6 cells in the 96-well format were infected with the diluted virus, then incubated for 24 hours at 37 °C with 5% CO2. GFP+ infected cells were counted using the IncuCyte S3 (Sartorius), which performed whole-well scans (4x) in phase contrast and green fluorescence (300 ms exposure). The IncuCyte GUI software (versions 2019B Rev1 and 2021B) was used for counting of fluorescent foci and counts were exported to Excel 2016. The average number of infected cells was calculated for three different pseudo-virus dilutions, and this number was multiplied by the dilution factor to determine the single cell focus units (sfu) per ml. The pseudo-virus titer was calculated as the mean sfu/ml from the three pseudo-virus dilutions.

### Neutralization assay
Neutralization assays were based on a previously published protocol[31]. In brief, all plasma samples utilized in this study were heat inactivated at 56 °C for 30 min, 10 fold diluted in DMEM [1% Penicillin-Streptomycin, 1% L-Glu, 5% FBS], and stored at 4 °C for further use. Pseudo-viral particles (600pfu/well± 30%) were incubated for one hour in a 96 well microtiter plate with two-fold diluted serum samples in DMEM [1% Penicillin-Streptomycin, 1% L-Glu, 5% FBS] ranging from 1:100 to 1:51200. Pseudo-virus particles were incubated in the absence of plasma as controls indicating 0% inhibition. After incubation, the serum/virus samples were transferred onto a confluent VeroE6 96 well plate. After a 24-h incubation, plates were fixed with 4% paraformaldehyde (PFA) and stored at 4 °C until readout. GFP+ infected cells were counted using an IncuCyte S3 (Sartorius) performing whole-well scans (4x) in phase contrast and green fluorescence settings (300 ms exposure). Automated segmentation and counting of fluorescent foci defined as green fluorescent protein GFP + -single cells was

**Neutralisation titers for S1/S2 and S2 specific BA.2.86pp mutants pre XBB.1.5 adapted vaccine**

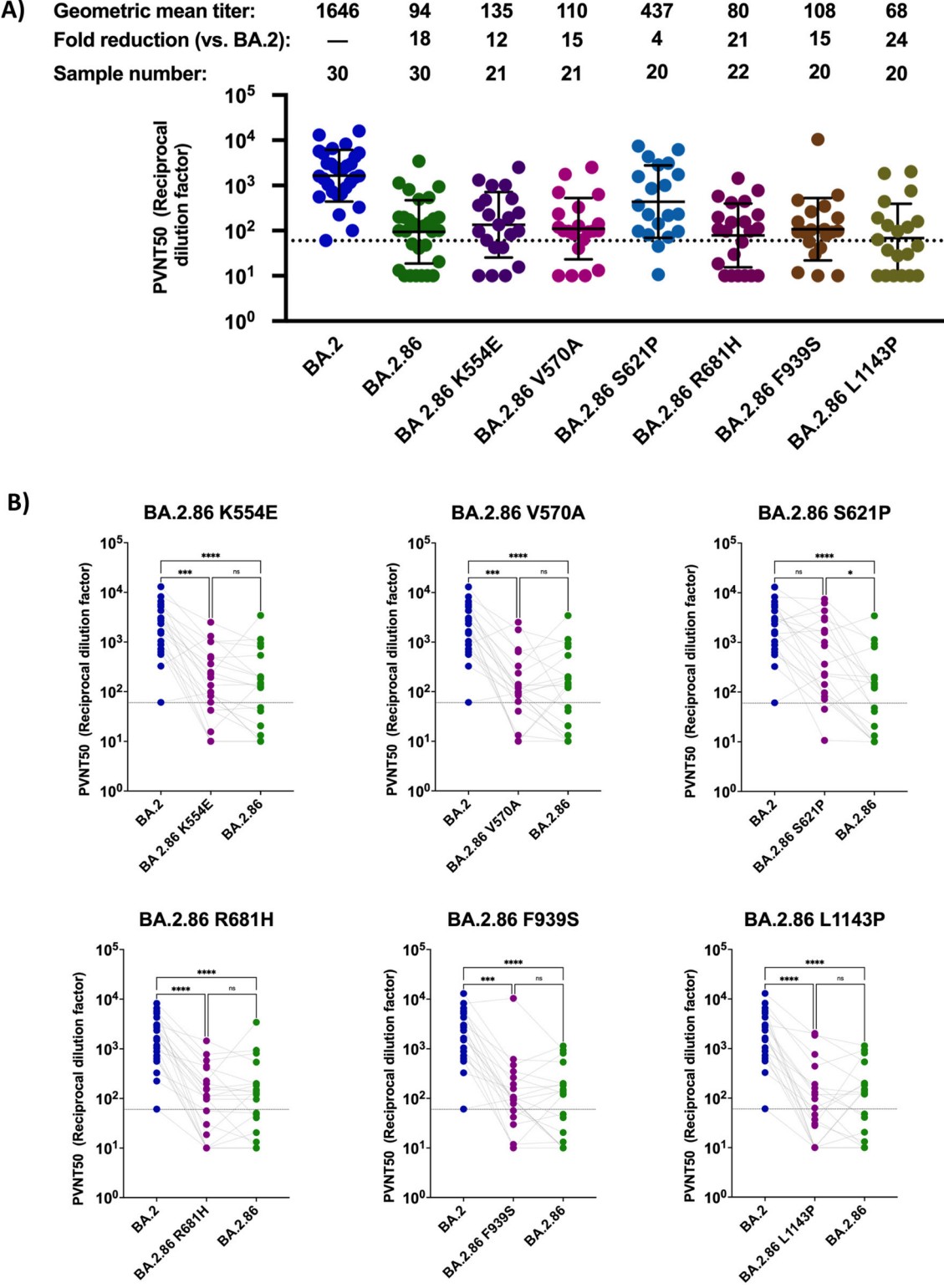

performed using the IncuCyte GUI software (versions 2019B Rev1 and 2021B). Pseudo-virus neutralization titer 50 (PVNT50) values were determined by a non-linear regression model. The lower limit of confidence (LLOC) was set at a PVNT50 of 50. Non-responders were defined as individuals with PVNT50 < 60 (dashed line). All PVNT50 < 10 are set at 10 for visualization purposes. Due to technical limitations

(the limited volume of plasma samples and the technical challenges associated with assaying 30 samples against ~40 different pseudo-viruses in duplicates), every pseudovirus was tested on at least 20 individual, randomly chosen plasma samples. These were arranged so that over 80% of the plasma originated from the same set of participants. This approach aimed to minimize variability due to differing

**Fig. 5 | Mapping mutations in the S1/S2 domain for effects of BA.2.86 neutralization efficiency in double boostered individuals. A** Neutralization assessment for particles pseudo-typed with mutations within the S1/S2 and S2 domain of BA.2.86. Each mutant shown contains a single mutation reverting the amino acid mutation in BA.2.86 to the corresponding amino acid within BA.2. Particles pseudo-typed with the indicated S proteins were preincubated for one hour at 37 °C with plasma dilutions from double boostered health care workers with non-adapted immunogens. The numbers of biological replicates corresponding to individual plasma samples are shown on top. Pseudo-virus neutralization titer 50 (PVNT50) was calculated using the least squares fit using a variable slope, using a four-parameter nonlinear regression model. The lower limit of confidence (LLOC) was set at a PVNT50 of 50. Non responders are defined as individuals below 60 (dashed line). All PVNT50 below 10 are set at 10 for visualization purposes. The assay was performed with negative controls to assess the virus input of each used pseudo-virus in the absence of plasma antibodies. Percentage of positive responders, geometric means, and fold change neutralization over BA.2pp are shown on top. Geometric mean standard deviation bars are shown in black. **B** Individual neutralization data for particles pseudo-typed with mutations within the S1/S2 and S2 domain of BA.2.86. Statistical significance was assessed by Friedman two sided nonparametric paired test (ns, $p > 0.05$; $*p \le 0.05$; $**p \le 0.01$; $***p \le 0.001$). Source data are provided as a Source Data file.

## Neutralisation titers for BA.2.86pp mutants post XBB.1.5 adapted vaccine

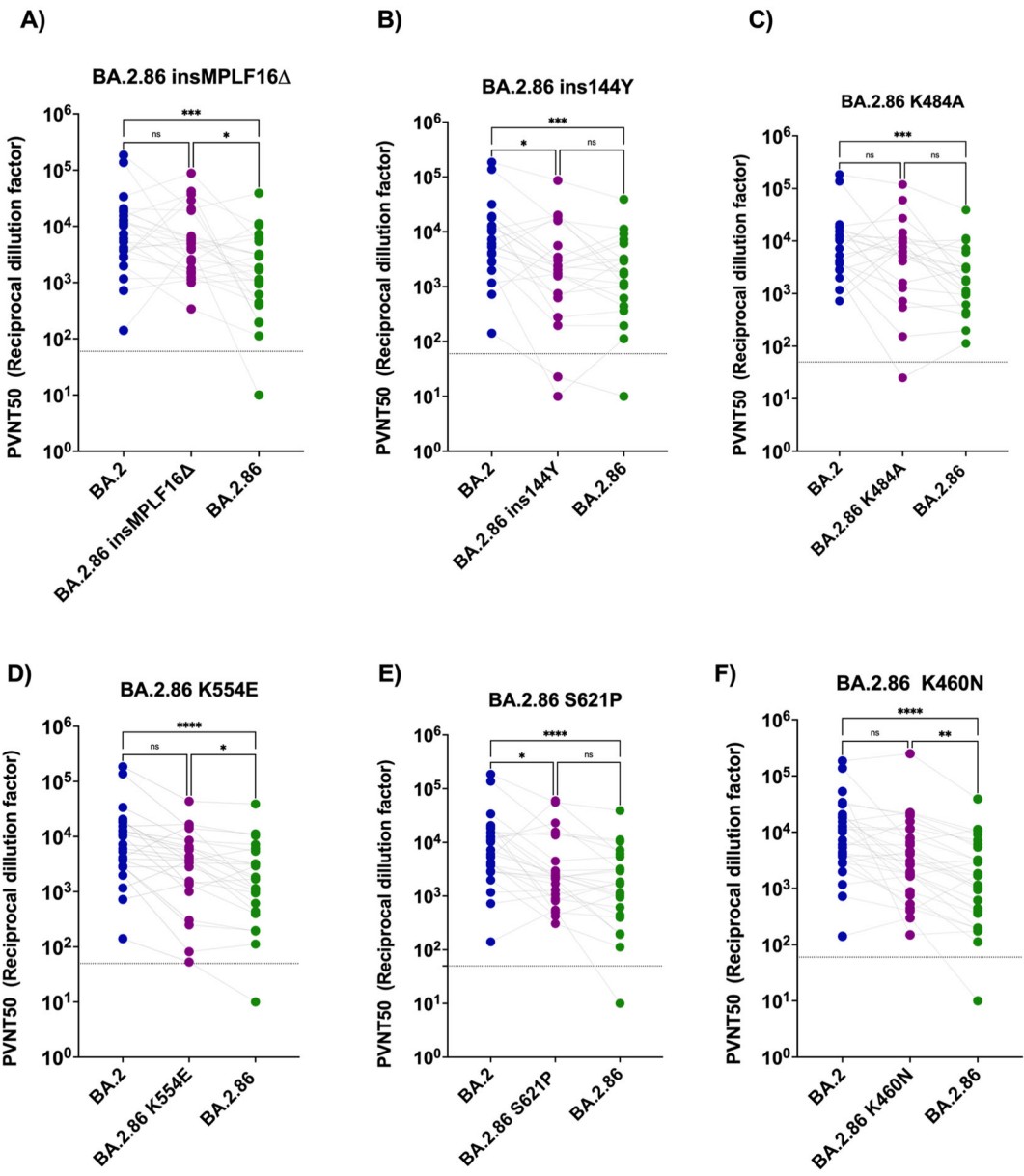

**Fig. 6 | Reversions insMPLF16Δ, K460N, K554E and to lesser extent insY144, K484A, and S621P enhance the neutralization efficiency of plasma samples post BNT162b2 XBB.1.5 vaccination. A–F** Neutralization assessment for pseudo-typed particles with plasma post BNT162b2 XBB.1.5 vaccination. Each mutant shown contains a single mutation reverting the amino acid mutation in BA.2.86 to the corresponding amino acid within BA.2. Particles pseudo-typed with the indicated S proteins were pre-incubated for one hour at 37 °C with plasma dilutions from double boostered health care workers. Pseudo-virus neutralization titer 50 (PVNT50) was calculated using the least squares fit using a variable slope, using a four-parameter nonlinear regression model. The lower limit of confidence (LLOC) was set at a PVNT50 of 50. Non responders are defined as individuals below this 60 (dashed line). All PVNT50 below 10 are set at 10 for visualization purposes. The assay was performed in technical duplicates and with negative controls to assess the virus input of each used pseudo-virus in the absence of plasma antibodies. Statistical significance was assessed by Friedman two sided nonparametric paired test (ns, $p > 0.05$; $*p \le 0.05$; $**p \le 0.01$; $***p \le 0.001$). Source data are provided as a Source Data file.

inherent potencies among plasma samples. All data are presented as numerical titers for each study participant in the Supplementary Data 2.

## Ethics committee approval

The collection of all plasma samples was approved by the research ethics committee of the Institutional Review Board of Hannover Medical School (8973_BO_K_2020). All donors provided written consent for plasma donation and use research purposes, consent for publishing identifiers such as sex and age was also acquired.

## Plasma samples

The number of participants within this analysis is n = 30. Median age is 45 years (interquartile range (IQR) 33–56). Male to female ratio is 1:2. Among these participants, 30% were vaccinated with three vaccine doses, 60% were vaccinated with four vaccine doses, and 10% were vaccinated with more than four doses. Ten participants (33.3%) were vaccinated with the bivalent WT/BA.4/5 vaccine. The median time in months since last recorded SARS-CoV-2 infection for the patients with known infections is 15 (IQR 13–18). The median number of months since the last known vaccination dates within our cohort is 13,5 months (IQR 11–22). All participants are part of the COVID-19 contact study, to monitor anti-SARS-CoV-2 immune responses in healthcare workers at Hannover Medical School (MHH). All participants donated blood directly prior to vaccination with 30 μg of the updated BNT162b2 Omicron XBB.1.5 vaccine (Raxtozinameran, BioNTech, Mainz, Germany) in September 2023 and were followed-up for another blood collection 15–16 days post vaccination[35]. Plasma was separated from collected blood and stored at −80 °C for long term storage and 4 °C for immediate use. Detailed information is provided in Supplementary Data 1.

## Viral genome copy number

To quantify the VSV genome copy numbers in samples, digital droplet PCR (ddPCR) was performed on the VSV-N gene. Supernatants containing viral particles were treated with nuclease muncher (Abcam) by incubating with 2 U/ml for >60 min at 37 °C before RNA isolation using the QIAamp Viral RNA Mini kit according to the manufacturer's instructions. For ddPCR, a one-step RT-ddPCR kit (Bio-Rad) was used combined with primers and probe detecting the VSV N (qRT VSV-N_F: ATGACAAATGGTTGCCTTTGTATCTACTT, qRT VSV-N R: ACGACCTT CTGGCACAAGAGGT, VSV-NcDNA probe: /56FAM/ACAGAGTGG/ZEN/ GCAGAACACAAATGCCT/3IABkFQ/) and, to exclude the presence of non-enveloped RNA in the sample, we used the human ACTB transcript (ACTBcDNA_F: GAGGAGCACCCCGTG, ACTBcDNA_R: GCCTGGATAG CAACGTAC, ACTBcDNA_probe:[HEX]CCCAGATCATGTTTGAGACCTT CAACACC[BHQ1]) PCR was done with a Bio-Rad C1000 Touch thermo cycler, encapsulation and read-out were done with the Bio-Rad QX200 system. Data analysis was carried out with Bio-Rad QX Manager Standard Edition (version 2.1.0.25).

## Statistical analysis

Statistical analysis was performed using GraphPad Prism 9.0 (Graph-Pad software). Neutralization titers were plotted as geometric mean titers. A two sided paired non-parametric Friedman test was performed for non-normally distributed data. $P$ values less than 0.05 were considered significant ns, $p > 0.05$; *$p \leq 0.05$; **$p \leq 0.01$; ***$p \leq 0.001$.

## Reporting summary

Further information on research design is available in the Nature Portfolio Reporting Summary linked to this article.

## Data availability

Values for neutralization titers and information on plasma donors are provided in Supplementary data 1 and 2. Data underlying the generation of the figures are provided as a source data file: Source Data. Source data are provided with this paper.

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

## Acknowledgements

This research was funded by the Helmholtz Association through Helmholtz Campaign COVIPA (KA1-Co-02) to L.C.-S. and EU-Partnering grant MCMVaccine (PIE-008) to L.C.-S. and S.P. Funding to L.C.-S and S.P. was also provided by the Ministry of Science and Culture of Lower Saxony, through the COFONI Network, flex fund projects 6FF22 and 10FF22. L.C.-S also obtained funding from the German Scientific Foundation (DFG) through grants EXC 2155 "RESIST"—Project ID 39087428 and FOR2830, project 5. G.M.N.B. and A.D-J. acknowledge funding from Ministry of Science and Culture of Lower Saxony (14-76103-184, COFONI Network, project 4LZF23), G.M.N.B. acknowledges funding by the European Regional Development Fund (ZW7-85151373), and A.D.-J. acknowledge funding by European Social Fund (ZAM5-87006761). We thank Ayse Barut, Yuliia Polianska, Inge Hollatz-Rangosch and Karina Watzke for expert technical assistance, Natascha Gödecke for support with biosafety compliance and Gert Zimmer (Institute of Virology and Immunology, Mittelhäusern, Switzerland) for providing the VSV pseudo-virus system. We also thank the CoCo Study participants for their support and the entire CoCo study team, especially Annika Heidemann and Luis Manthey, for technical and logistical support.

## Author contributions

Conceptualization: L.C.-S.; Methodology: L.C.-S.; N.B.; Investigation: N.B., T.L., H.M., K.M., H.J., S.S. U.K., U.R.; Formal analysis: N.B., S.S., W.B., U.R., A.D-J., G.M.N., L.C.-S.; Resources: S.P., M.H., A.C., M.V.S., A.D.-J., and G.M.N.; Funding acquisition: L.C.-S.; Writing original draft: N.B.; Writing review & editing: all authors.

## Funding

## Competing interests

L.C.-S. served as an advisor to Sanofi unrelated to this work. G.M.N.B. served as advisor for Moderna unrelated to this work, A.D.-J. served as an advisor for Pfizer unrelated to this work, S.P. and M.H. conducted contract research (testing of vaccinee plasma for neutralizing activity against SARS-CoV-2) for Valneva unrelated to this work. S.P. served as advisor for BioNTech, unrelated to this work. H.J. served as advisor on COVID neutralization assays for WHO and CEPI, unrelated to this work. The other authors declare no competing interests.
