## [Peer Review File · Nature Communications]

Reverse mutational scanning of SARS-CoV-2 spike BA.2.86 identifies epitopes contributing to immune escape from polyclonal seraEditorial Note: Parts of this Peer Review File have been redacted as indicated to remove third-party material where no permission to publish could be obtained.

REVIEWER COMMENTS

Reviewer #1 (Remarks to the Author):

Overall, this is an interesting study that dissects the contributions of mutations in BA.2.86 to serum antibody escape. I consider this to be an important and timely study: I found it interesting, and others in the field will too. I strongly support its publication in Nature Communications with just the following minor suggested revisions.

MINOR COMMENTS:

- The legend to Fig 1A should clearly specify that XBB.1.5 and EG.5.1 have other mutations relative to BA.2 that are not shown, as only sites also mutated in BA.2.86 are indicated in the schematic.

- Figure 3A and 5A indicate the fold change as -18, etc, whereas Figure 4A indicates it as 18x. Be consistent in how this is labeled.

- As a minor comparison point to the Dadonaite et al deep mutational scanning study cited by the authors, in fact many of the mutations have a similar effect in that study as the current one. In particular, go to this website (https://dms-vep.org/SARS-CoV-2_XBB.1.5_spike_DMS/htmls/summary_overlaid.html) and click the "floor escape at zero" option at the bottom to "false". If you mouse over the top heatmap, you can then see that T356K is an escape mutation, K460N reversion is sensitizing, V483del is an escape mutation, P486F reversion is sensitizing, and S621P reversion is sensitizing. This point does not necessarily need to be called out in the current study, but it is interesting that both studies are given largely consistent results on the effects of these mutations.

- Please provide the actual numerical titers for each individual sera against each mutant as a supplementary table. This will be useful for others who want to re-analyze the data.

- Lines 160-162: it is confusing when authors switch between "more efficient neutralization" and "reduced neutralization efficiency". Authors state K356T is ~3 fold more efficiently neutralized than BA.2.86 which is the same as 6 fold reduction in neutralization as indicated in Fig.4A but stating it that way gives a confusing impression that change in neutralization is significantly more pronounced with K356T compared say K460N when in reality the former has 6.4 (180/28) and the latter has 6.8 (180/26.3) fold reduction in neutralization.

- Can authors explain why the number of sera samples used for neutralization of different reverse mutants is not always the same? Maybe some sera volumes ran out and could not be tested against all mutants (?), but this should be mentioned including if it was the same set of sera or not for different mutants. Because different sera have different inherent potencies, this could have some effect on the results.

- The title of the study is misleading in that the authors never do reverse mutagenesis in JN.1 background, it is done in the BA.2.86 background.
- Line 94: Should state XBB.1.5 not XBB.21.5
- Line 96: Should state K554E not K544E
- Fig.3B: Some titles lack "6" in BA.2.86

Reviewer #2 (Remarks to the Author):

The manuscript by Bdeir et al. describes the impact of individual mutations on immune evasion in BA.2.86 and its derivative JN.1 (BA.2.86 + S455L) lineage. They employed a reverse mutational scanning approach where single mutations out of 33 BA.2.86 ones were returned to BA.2 and found that 16insMPLF in NTD, K356T, N460K, V483Δ, A484K, F486P in RBD, and P621S in S1/S2 contributed to escape from Wuhan vaccinated sera. In addition, 16insMPLF, Δ144Y, E544K, P621S, and A484K were involved in the case of the XBB.1.5 booster. These data are important to understand the evolution of SARS-CoV-2. However, information is very limited, and it is difficult to deeply discuss it.

Viral transmissibility is mainly determined by infectivity and immune escapability, and ACE2 affinity and structural stability are key factors for infectivity. Immune evasion is sometimes achieved at the expense of ACE2 affinity and structural stability. In this case, additional mutations compensate the compromised affinity and stability. From these perspectives, the reviewer requests that the change in serum antibody binding, ACE2 affinity and structural stability be examined in the same setting of vaccine evasion and that the overall evolution of the virus be evaluated.

Minor concerns,

As logic for the advantage of reverse mutational scanning, this manuscript says that mutating one out of 33 epitopes on an ancestral background may only marginally decrease the serum neutralizing activity if some among the remaining 32 epitopes are recognized by independent antibody clones. If serum neutralization is saturated, I think that is correct. However, the neutralization of BA.2 should be weaker than that of Wuhan and may not be saturated. It is better to compare with standard add-on mutational scanning to appeal this approach.

E544K should be E554K in summary and other section.

REVIEWER COMMENTS

Reviewer #1 (Remarks to the Author):

Overall, this is an interesting study that dissects the contributions of mutations in BA.2.86 to serum antibody escape. I consider this to be an important and timely study: I found it interesting, and others in the field will too. I strongly support its publication in Nature Communications with just the following minor suggested revisions.

We thank the author for his positive remarks on our work.

MINOR COMMENTS:

Remark: The legend to Fig 1A should clearly specify that XBB.1.5 and EG.5.1 have other mutations relative to BA.2 that are not shown, as only sites also mutated in BA.2.86 are indicated in the schematic. Answer: We have altered the figure legend accordingly. Please see lines 467-468

Remark: Figure 3A and 5A indicate the fold change as -18, etc, whereas Figure 4A indicates it as 18x. Be consistent in how this is labeled. Answer: All figures have been altered for consistence.

Remark: As a minor comparison point to the Dadonaite et al deep mutational scanning study cited by the authors, in fact many of the mutations have a similar effect in that study as the current one. In particular, go to this website (https://dms-vep.org/SARS-CoV-2_XBB.1.5_spike_DMS/htmls/summary_overlaid.html) and click the "floor escape at zero" option at the bottom to "false". If you mouse over the top heatmap, you can then see that T356K is an escape mutation, K460N reversion is sensitizing, V483del is an escape mutation, P486F reversion is sensitizing, and S621P reversion is sensitizing. This point does not necessarily need to be called out in the current study, but it is interesting that both studies are given largely consistent results on the effects of these mutations.

Answer: We thank the reviewer for bringing this interesting piece of work to our attention.

Remark: Please provide the actual numerical titers for each individual sera against each mutant as a supplementary table. This will be useful for others who want to re-analyze the data.

Answer: An additional supplementary file displaying individual titers has been provided, Supplementary file 2. This supplementary files contains the neutralization titers of participants' plasma before and after vaccination against every mutant pseudovirus. Furthermore, all data for post-vaccination titers are now shown in a visualized form as Supplementary figure 2. A reference to this data has been added to the manuscript, lines 176-178. Moreover, we noticed we overlooked a 10 fold pre-dilution of the samples; this has now been adjusted and in consequence, all figures display a 10-fold increase in nominal titers. Hence, the limit of detection has also been adjusted, Lines 411-413.

Remark: Lines 160-162: it is confusing when authors switch between "more efficient neutralization" and "reduced neutralization efficiency". Authors state K356T is ~3 fold more efficiently neutralized than BA.2.86 which is the same as 6 fold reduction in neutralization as indicated in Fig.4A but stating it that way gives a confusing impression that change in neutralization is significantly more pronounced with K356T compared say K460N when in really the former has 6.4 (180/28) and the latter has 6.8 (180/26.3) fold reduction in neutralization.

Answer: We apologize for the confusion. The text has been edited in order to represent fold reduction over BA.2 and not fold increase against BA.2.86 for K356T, lines 155-156.

Remark: Can authors explain why the number of sera samples used for neutralization of different reverse mutants is not always the same? Maybe some sera volumes ran out and could not be tested against all mutants (?), but this should be mentioned including if it was the same set of sera or not for different mutants. Because different sera have different inherent potencies, this could have some effect on the results.

Answer: The discrepancy in the number of plasma samples assayed for each virus can be attributed to two main factors: the limited volume of plasma and the technical challenges associated with assaying all 30 patients against nearly 40 different pseudovirus species in duplicates. Consequently, sample numbers in our assays were randomly distributed to ensure at least 20 individual samples within each group. For the different virus mutants, results were arranged so that over 80% of the plasma originated from the same set of patients. This approach aimed to minimize variability due to differences in neutralization potencies among plasma samples. Further details, including individual neutralization titers for each volunteer against each virus, are provided in the new supplementary file 2. The materials and methods section has been updated to reflect this information, 412-419

Remark: The title of the study is misleading in that the authors never do reverse mutagenesis in JN.1 background, it is done in the BA.2.86 background. **Answer:** JN.1 has been removed from the title.

Remark: Line 94: Should state XBB.1.5 not XBB.21.5. **Answer:** This has been corrected lines 89.

-Remark: Line 96: Should state K554E not K544E. **Answer:** This has been corrected lines 91.

Remark: Fig.3B: Some titles lack "6" in BA.2.86. **Answer:** This has been corrected in Fig3B

Reviewer #2 (Remarks to the Author):

The manuscript by Bdeir et al. describes the impact of individual mutations on immune evasion in BA.2.86 and its derivative JN.1 (BA.2.86 + S455L) lineage. They employed a reverse mutational scanning approach where single mutations out of 33 BA.2.86 ones were returned to BA.2 and found that 16insMPLF in NTD, K356T, N460K, V483Δ, A484K, F486P in RBD, and P621S in S1/S2 contributed

to escape from Wuhan vaccinated sera. In addition, 16insMPLF, Δ144Y, E544K, P621S, and A484K were involved in the case of the XBB.1.5 booster. These data are important to understand the evolution of SARS-CoV-2. However, information is very limited, and it is difficult to deeply discuss it.

Viral transmissibility is mainly determined by infectivity and immune escapability, and ACE2 affinity and structural stability are key factors for infectivity. Immune evasion is sometimes achieved at the expense of ACE2 affinity and structural stability. In this case, additional mutations compensate the compromised affinity and stability. From these perspectives, the reviewer requests that the change in serum antibody binding, ACE2 affinity and structural stability be examined in the same setting of vaccine evasion and that the overall evolution of the virus be evaluated.

Answer: We thank the reviewer for their incisive remarks. We considered them thoroughly, and we think that it is highly unlikely that our results can be explained by artefacts. We provide several lines of reasoning for this conclusion:

All of our analyses were performed on cells infected in the presence of serial plasma dilutions, but the number of infected cells was always referenced to a control well, where the same mutant was used in absence of antibodies. Hence, the values shown in our assays correspond to the plasma dilutions that led to a 50% reduction in infectivity relative to plasma-untreated, infected cells. Therefore, our results do not define an absolute change in infectivity, but one that is relative to a control group. Hence, an overall reduction in ACE2 binding and cell entry would affect the virus both in the presence of antibodies and in their absence, which should not be necessarily reflected in our data.

Nevertheless, we agree that an assessment for ACE-2 binding and infectivity is essential to exclude that virus neutralization is altered due to a massive loss of infectivity of the pseudovirus. We observed this phenomenon in a recent study where we performed a similar library assessment of Omicron mutations on the Wuhan background and noticed several point mutations which compromised the infectivity of the virus stock by a factor of 100x or more (Katzmarzyk et al. Front. In Immunol. 2023). Two among these mutants showed indeed a 4-7 fold loss in neutralization capacity (Katzmarzyk et al. Front. In Immunol. 2023 supplementary Fig. 1) and we showed in the same paper that inactivated virus particles may act as a sponge for antibodies, thus marginally affecting the neutralization assay accuracy (Katzmarzyk et al. Front. In Immunol. 2023, supplementary Fig. 2C). Therefore, if our viruses were to display a loss in infectivity of 100-folds or more, this alone would have been sufficient to significantly reduce the nominal neutralization titer, as an artifact of the conditions, where defective virus particles compete with the infective ones for antibody binding. This, however, cannot explain the increase in titer relative to BA.2.86 that was observed in the point mutants in figures 3-6, because a loss of infectivity would result in a reduction of the same titers.

Regardless, we have investigated the mutations in context of their ability to bind to and enter the target Vero or Calu-3 lung cells. This was not the scope of the present manuscript, and hence it was initiated in parallel with the current manuscript and has been published in collaboration with a partner lab as a more thorough characterization of our mutants (Zhang et.al PMID: 38194966 DOI: 10.1016/j.cell.2023.12.025). Since that study was not accepted at the time of this submission, we did not cite and discuss it. The related paper is now appropriately cited and the results discussed in the

context of our new data 181-188 and 269-278. We show below the relevant figure from that publication (Suppl. Fig 3), where the upper row displays the entry into Vero cells from each mutant in the library, while the lower row the entry into Calu3 cells. Only the results of entry into Vero cells are relevant for this discussion, because Veros were used in the neutralization assay.

Zhang et.al (Supplementary figure 3)

[REDACTED]

Several mutants display a decrease in infectious titer in Vero cells, but this pattern does not match the pattern we reported from the neutralization assay. Nevertheless, the mutant Q493R, which shows the biggest drop in infectivity, shows also a (non-significant) 2-fold reduction in neutralization titers relative to BA.2.86 (Fig. 4). Hence, we performed a ddPCR of viral genomes in the mutant virus stocks to establish their relationship to infectious titers and identify if the number of physical virus particles varied among the mutants that showed significant differences in infectivity or in plasma neutralization. The results are shown in the rebuttal figure 1 below and in the supplementary figure 3 Lines: 181-188 and 279-286.

Rebuttal Figure 1:

[REDACTED]

All pseudoviruses were within the variation of genome copies per infectious particle observed in the naturally occurring BA.2 and BA.2.86 variants. Therefore, the number of non-infectious particles that may have acted as sponges was rather low. As we showed in a previous publication, a relationship between infectious and non-infectious particles that results in an artefactual 50% reduction in neutralization titer would need to be at least 10:1 (see Suppl. Fig 3 from Kazmarzyk et al. 2023) and below.

Katzmarzyk et. al (Supplementary figure 2)

Since in the Fig. S3 (Rebuttal figure 1) we observed values much lower than that, we are not concerned about artefacts in plasma titration. Therefore, to the best of our understanding, the reported increase in the neutralization titers of the mutants relative to BA.2.86 is not affected by cell infection properties.

Minor concerns,

As logic for the advantage of reverse mutational scanning, this manuscript says that mutating one out of 33 epitopes on an ancestral background may only marginally decrease the serum neutralizing activity if some among the remaining 32 epitopes are recognized by independent antibody clones. If serum neutralization is saturated, I think that is correct. However, the neutralization of BA.2 should be weaker than that of Wuhan and may not be saturated. It is better to compare with standard add-on mutational scanning to appeal this approach.

Answer: We are not sure that serum antibodies can ever be considered saturated in the chemistry-oriented sense that a solution is saturated, but the population that we assayed was repeatedly vaccinated, boosted and exposed to infections. Hence, BA.2 specific titers were almost as high as the Wuhan-specific titers (see fig 2). Moreover, in a previous study we showed that forward mutational scanning results in rather weak decreases in neutralization titers (Katzmarzyk et al. Front in Immunol. 2023). We discuss these points in the discussion (lines 203-213).

E544K should be E554K in summary and other section.

Answer: This has been corrected across the entire manuscript.

REVIEWERS' COMMENTS

Reviewer #1 (Remarks to the Author):

The authors have satisfactorily addressed the comments, and I support publication of the manuscript.

Reviewer #2 (Remarks to the Author):

The data from Zhang et.al (Supplementary figure 3C and D) is the one I requested. In the case of Omicron BA.1., each mutation contributed harmoniously to immune escape, ACE2 affinity, and spike stability (e.g. PMID: 35988543). I expected a similar evolution in BA. 2.86, and the infection and spike stability/expression data allow us to discuss it. It is unfortunate that these data cannot be included in this paper, but discussion with published data is also acceptable. In the RBD, it can be assumed that V483del achieves escape at the expense of infectivity, but R493Q compensates for infectivity. One question is why insMPLF16 is not highlighted. This insertion is involved in both immune evasion and infectivity (probably due to stability). It is unusual that the NTD mutation achieves both advantages with high impact. It is better to mention this unique insertion mutation in the discussion. Other concerns have been adequately addressed.

REVIEWERS' COMMENTS

Reviewer #1 (Remarks to the Author):

The authors have satisfactorily addressed the comments, and I support publication of the manuscript.

Answer: We thank the author for his decision.

Reviewer #2 (Remarks to the Author):

The data from Zhang et.al (Supplementary figure 3C and D) is the one I requested. In the case of Omicron BA.1., each mutation contributed harmoniously to immune escape, ACE2 affinity, and spike stability (e.g. PMID: 35988543). I expected a similar evolution in BA. 2.86, and the infection and spike stability/expression data allow us to discuss it. It is unfortunate that these data cannot be included in this paper, but discussion with published data is also acceptable. In the RBD, it can be assumed that V483del achieves escape at the expense of infectivity, but R493Q compensates for infectivity. One question is why insMPLF16 is not highlighted. This insertion is involved in both immune evasion and infectivity (probably due to stability). It is unusual that the NTD mutation achieves both advantages with high impact. It is better to mention this unique insertion mutation in the discussion. Other concerns have been adequately addressed.

Answer: We have added a short discussion referencing this unique feature within our discussion section of the manuscript.

Additional Considerations to both Reviewers :

We have made some corrections and modifications to our manuscript since the last submission.

The main reason for this correction was that we noticed that some samples showed lower antibody titers upon the booster vaccination, mainly for titers against WT-Wuhan, but on occasion against other variants as well. Since booster vaccinations should increase titers, rather than decrease them, we considered this a potential technical issue that we decided to rigorously address. These records were validated by repeating side-by-side titrations.

Contrary to numerous previous studies, where the titrations of before and after vaccination were in the focus of the scientific question and thus performed in parallel, the before and after comparison was not a major focus of this study. Namely, the main aim of our study was to compare the titers of individual single point mutants relative to BA.2.86.

Therefore, the before/after challenge titrations were performed not only on separate days, but on entirely separate weeks, which resulted in some batch effects, which were mostly identified in figure 2 for WT-Wuhan pseudovirus titrations. We performed a new set of side-by-side analyses of before and after vaccinations, and we have revised the results according to this new experimental evidence. The curated results reflect more closely evidence published by others and demonstrate a more pronounced escape of JN.1 both before and after the XBB1.5 booster.

We have also repeated results for other naturally occurring variants and point mutants, where a similar drop in titer was observed and we have normalized all titration curves for consistency. Hence, we have reanalyzed all datasets and comparisons, as shown in figures 3-6 to ascertain if this results in any differences in significance.

Some adjustments of geometric means occurred, but the results remained overall similar to the previous ones. We notice no major differences in titer reductions over BA.2 for mutants that were identified as contributing to neutralization escape pre boost. However, some p values changed, and this is indicated in the figure panels in the attached file.

- For the pre-booster results, BA.2.86 insMPLF16 remained significant, but the significance threshold changed from $p < 0.01$ to $p < 0.05$.
- The RBD epitope T356K remained significant, as did the P621S (both at the same level).
- Other results remained non-significant, with the exception of the $\Delta 483V$ mutant from the RBD, which was now also identified as significantly different from BA.2-86 at the $p < 0.05$.
- Two epitopes that showed a significant difference in neutralizing titers relative to BA.2.86 upon boosting (insMPLF16 and E554K) remained significant, but their significance changed from $p < 0.01$ to $p < 0.05$.
- Additionally, the N460K epitope was now identified as significant. Finally, the revised values for the S621P mutant showed no significant difference in titers relative to BA.2.86.
- All other differences in values upon file curation affect descriptive statistics, but not the statistical significance and are indicated in the slides.

We have provided the manuscript in a track change format to indicate what has changed from the previous version and we have provided a power point file showing the side by side comparison of all figures before and after modification for better clarity.

We believe that this effort has diminished the noise in our dataset and thus increased the quality of the data.